# Anisortopic Modeling of Hydraulic Fractures Height Growth in the Anadarko Basin

Ahmed Merzoug [1,*] , Abdulaziz Ellafi [1,*] , Vamegh Rasouli [2] and Hadi Jabbari [1]

1 Department of Petroleum Engineering, University of North Dakota, Grand Forks, ND 58202, USA
2 Department of Petroleum Engineering, University of Wyoming, Laramie, WY 82071, USA
* Correspondence: ahmed.merzoug@und.edu (A.M.); abdulaziz.ellafi@und.edu (A.E.)

**Abstract:** Correct estimation of hydraulic fracture height growth is a critical step in the design of Hydraulic Fracturing (HF) treatment, as it maximizes the reservoir stimulation and returns on investment. The height of the fractures is governed by several in situ conditions, especially stress variation with depth. The common workflow to estimate stress is by building the mechanical earth model (MEM) and calibrating it using the Diagnostic Fracture Injection Test (DFIT). However, DFIT interpretation is a complex task, and depending on the method used, different results may be obtained that will consequently affect the predicted hydraulic fracture height. This work used the tangent and compliance methods for DFIT interpretation, along with isotropic and anisotropic stress profiles, to estimate the HF height growth using numerical modeling in a 3D planar HF simulator. Data from two wells in the Anadarko Basin were used in this study. The predicted height was compared with microseismic data. The results showed that even though the tangent method fits better to the isotropic stress profile, HF did not match with the microseismic data. On the contrary, the anisotropic stress profile showed a good match between the compliance DFIT model and the microseismic events. Based on the discussions presented in this study, the validity of the DFIT interpretation is debatable, and when the formations are anisotropic, the isotropic model fails to correctly estimate the minimum stress profile, which is the main input for the estimation of the fracture height. This is in addition to the fact that some researchers have questioned the use of the tangent method in low-permeability formations.

**Keywords:** multistage hydraulic fracturing; anisotropic formation; DFIT; Anadarko Basin

## 1. Introduction

The extraction of unconventional resources was made possible thanks to the combination of horizontal drilling and multistage HF technologies. Operators in the United States try to secure full development rights by placing the first horizontal well in a field followed by other wells in an attempt to increase production. The process is a combination of long laterals with millions of square feet of contact area filled with proppant in lower permeability unconventional reservoirs [1–7].

The length, height, and width characterize the fracture geometry. Where the length represents the extent of the fracture in the reservoir and the drainage area, fracture height is the vertical coverage of the HF treatment. For an appropriate design of HF jobs, fracture height estimation is essential for optimum well placement and geometry estimation [8,9].

Several factors can influence hydraulic fracture growth, such as stress contrast, in situ stress gradient, elastic properties contrasts with different layers, discontinuities, and natural fractures [10,11]. Fisher and Warpinski [12] assert, using real data, that stress variation with depth controls HF growth. Ganpule et al. [13], Pandey et al. [14] and Hryb et al. [15] emphasized the importance of building a mechanical earth model (MEM) for accurate stress estimation and modeling for HF treatment. The process of building MEM is not a trivial task, as the model contains several degrees of freedom which require extensive data

collection and calibration with in situ stress measurements. Diagnostic fracture injection tests have been used for in situ stress measurements and MEM calibration; however, the interpretation of DFIT is subjective, and different methods may be used, which result in different values with possibly more than a 1000 psi difference [16–18]. The calibrated MEM using DFIT is then used as the input into the numerical simulators to estimate the fracture geometry and optimizing the fracture landing zone for a better return on investment and optimum design [9,19]. Thus, erroneous predictions can result from these biased MEMs.

Oyarhossein and Dusseault [20] used numerical modeling to investigate the effect of different parameters on fracture height growth. They noted that stress, elastic properties, and injected fluid properties are the most influential parameters on fracture height. In this study, we keep the elastic properties and the injected fluid properties the same for all cases. The comparison is based on the stress estimation approaches. This is supported by the work of Ben Naceur & Touboul [10]. They argue the most influential parameter on the height growth is stress contrast, whereas the elastic moduli only slow the fracture propagation.

The current study demonstrates the importance of using an anisotropic stress estimation model in anisotropic formations using a case study in the Anadarko Basin. The minimum stress profile is the key input for fracture height estimation. The case study belongs to two wells from Woodford and Meramec's plays of the Anadarko Basin.

## 2. Problem Statement

This work aims at modeling the HF height growth in two wells drilled in Woodford and Meramec formations, referred to as Well #1 and Well #2, respectively. The fracture height propagation noted in the microseismic data and the stress estimation from DFIT were not in good agreement. The aim was to explain the discrepancy between stress estimation and the geometry estimated from microseismic data. Figure 1 illustrates the location of the two wells compared with each other. The completion design pumped in each well is reported in Table 1.

The timeline for drilling and the production events of these wells is illustrated in Table 2. Well #1 was hydraulically fractured first, followed by Well #2 after two weeks. The microseismic data shows that the induced fractures from Well #1 propagated upward toward Well #2, and the fracture from Well #2 propagated downwards towards Well #1. Two stress models were built and calibrated to model the HF growth based on the microseismic data.

**Table 1.** Completion design and pumped treatment in Well #1 and Well #2.

| Parameters | Well #1 | Well #2 |
|---|---|---|
| Lateral length | 10,000 ft | 10,000 ft |
| Stage number | 40 | 36 |
| Cluster/stage | 4 | 5 |
| Cluster spacing | 60 ft | 50 ft |
| Perfs per cluster | 4 | 15 |
| Perf diameter | 0.4 in | 0.4 |
| Total slurry volume | 694,218 bbl | 489,021 bbl |
| Average rate | 100 bpm | 100 bpm |
| Proppant mass and type | 18,966,860 lb (40% 100 mesh, 20% 30/50 premium, 20% 40/70 premium, and 20% 40/70 premium) | 24,675,997 lb (10% 100 mesh, 55% 30/50 premium, 35% 20/40 premium) |

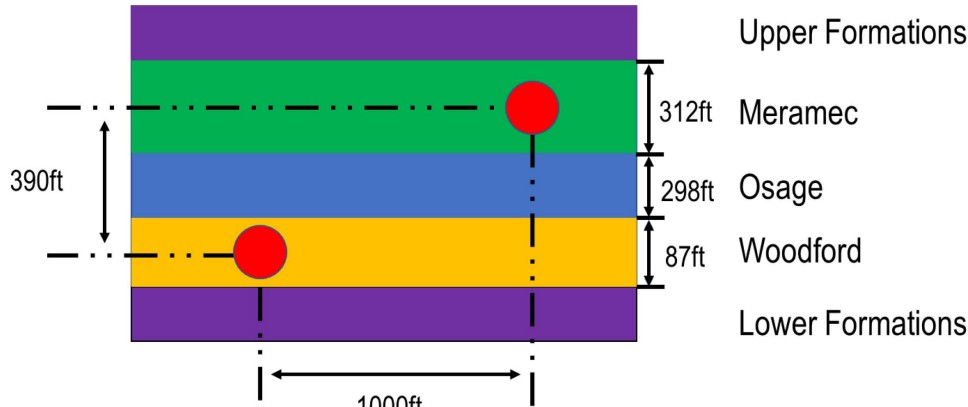

**Figure 1.** A gun barrel schematic of the location of the two study wells. The well in the Woodford is Well #1 and the well in the Meramec is Well #2.

**Table 2.** Chronology of drilling and production events in the two wells of this study.

| Time (Days) | Event |
| --- | --- |
| 0 | Both wells are drilled, and Well #1 is hydraulically fractured |
| 15 | Well #2 is hydraulically fractured |
| 90 | Well #1 is put on production |
| 100 | Well #2 is put on production |

## 3. Stress Estimation

The simplest approach for the stress-strain relationship model assumes the rock is isotropic, meaning that the rock properties are similar in different directions. The isotropic symmetry model is widely applied to estimate the rock's elastic properties, due to its simplicity. However, many studies performed in shales proved that the assumption of isotropy would result in significant errors in the estimation of rock properties and stress profiles [21–25]. Most rocks exhibit anisotropic behavior to some extent. The anisotropy can result from the laminated nature of some formations, such as shale or the existence of microcracks [26]. Shales show higher elastic anisotropy values that cannot be accounted for in an isotropic model [21,22,24,25,27,28].

Shales, which are laminated rocks, can be approximated as a transverse isotropic medium (TI), which assumes that the properties are symmetrical around one axis, which is normal to a plane of isotropy [26]. It implies that rock properties are only equal in the same plane but different in the other directions. This implies that two samples in perpendicular and parallel, with respect to the lamination, are required to estimate two Young's moduli, two Poisson's ratios, and one shear modulus to fully characterize its elastic properties [21–23,26]:

The three principal stresses $\sigma_v$, $\sigma_h$, and $\sigma_H$ can be calculated as follows:

$$\sigma_v(z) = \int_0^z \rho(z)g\,dz \tag{1}$$

where: $\sigma_v(z)$ is the total vertical stress at depth $z$, $\rho(z)$ is the density value at depth z, and $g$ is the acceleration due to gravity.

For an isotropic rock, the poroelastic model is commonly used to estimate the horizontal stresses as [29,30]:

$$\sigma_h - \alpha P_p = \frac{v}{1-v}(\sigma_v - \alpha P_p) + \frac{E}{1-v^2}\varepsilon_h + \frac{Ev}{1-v^2}\varepsilon_H \tag{2}$$

$$\sigma_H - \alpha P_p = \frac{v}{1-v}\left(\sigma_v - \alpha P_p\right) + \frac{E}{1-v^2}\varepsilon_H + \frac{Ev}{1-v^2}\varepsilon_h \tag{3}$$

where: $\sigma_h$ is the total minimum horizontal stress, $\sigma_H$ is the total maximum horizontal stress, $\alpha$ is the biot coefficient, $\varepsilon_h$ is the minimum tectonic strain, and $\varepsilon_H$ is the maximum tectonic strain.

For a TI medium, the poroelastic model is written as [29,30]:

$$\sigma_h - \alpha P_p = \frac{E_h}{E_v}\frac{v_v}{1-v_h}\left(\sigma_v - \alpha P_p\right) + \frac{E_h}{1-v_h^2}\varepsilon_h + \frac{E_h v_h}{1-v_h^2}\varepsilon_H \tag{4}$$

$$\sigma_H - \alpha P_p = \frac{E_h}{E_v}\frac{v_v}{1-v_h}\left(\sigma_v - \alpha P_p\right) + \frac{E_h}{1-v_h^2}\varepsilon_H + \frac{E_h v_h}{1-v_h^2}\varepsilon_h \tag{5}$$

Note that the assumptions of these equations are that the earth is elastic, and the overburden is applied instantaneously. The sources of horizontal stresses are overburden and tectonic stresses. This approach has been criticized for having many degrees of freedom and the assumption that stress is applied instantaneously [31].

Another stress estimation approach is the frictional limit theory, which states that the earth's crust has limited strength. This limit constrains the stress state at any depth [31]. The crust limit is generally defined as the frictional strength of faults. The relationship between the maximum and minimum principal stresses is expressed as [23]:

$$\frac{\sigma_1{}'}{\sigma_3{}'} = \frac{\sigma_1 - P_p}{\sigma_3 - P_p} = \left[\left(\mu^2 + 1\right)^{\frac{1}{2}} + \mu\right]^2 \tag{6}$$

where: $\sigma_1{}'$ is the effective maximum principal stress, $\sigma_3{}'$ is the effective minimum principal stress, $\sigma_1$ is the total maximum principal stress, $\sigma_3$ is the total minimum principal stress, and $\mu$ is the friction coefficient

Acknowledging the assumption of stress estimation from logs for TI medium, different data sources have been used to build a mechanical earth model and constrain it using logs, core data, DFITs, and microseismic events. Note that the Drilling Spacing Unit (DSU) field is located in the Anadarko Basin, where the stress regime is strike-slip with a relative stress magnitude of 1.6 [32]. Both isotropic and anisotropic stress models were used for comparison and matching data reported from microseismic data.

## 4. Methodology

In this study, for comparison purposes, both isotropic and anisotropic stress models were built and calibrated using different interpretations of DFIT. The isotropic stress model was calibrated with the tangent method [33]. Whereas the anisotropic model was calibrated with the compliance method [17], supported by the first separation of the Gdp/dG plot reported by Nadimi et al. [34] and Pandey and Agreda [35]. The HF geometry was estimated using numerical simulations for both constructed stress models and compared with microseismic reported data.

The wireline logs from a vertical reference well at a 1500 ft distance from the hydraulically fractured wells were used in this study. A gamma-ray log was used to mark the formation tops. Density and sonic logs were used to construct the MEM. The operators acquired core data to estimate the mechanical properties of the rock using lab testing. The operator only cored intervals across the Meramec and the Osage formations; so, the Woodford core mechanical properties were obtained from published data. The stress regime was identified as a strike-slip based on the North America Stress Map [32]. The stress was estimated using the extended Eaton method and calibrated using DFIT from the DSU and nearby wells. The microseismic data was recorded downhole from four monitoring wells using arrays of 3 component geophones. The treatment design (fluid and proppant) was

also reported in this section. The workflow used in this study is depicted in the diagram of Figure 2.

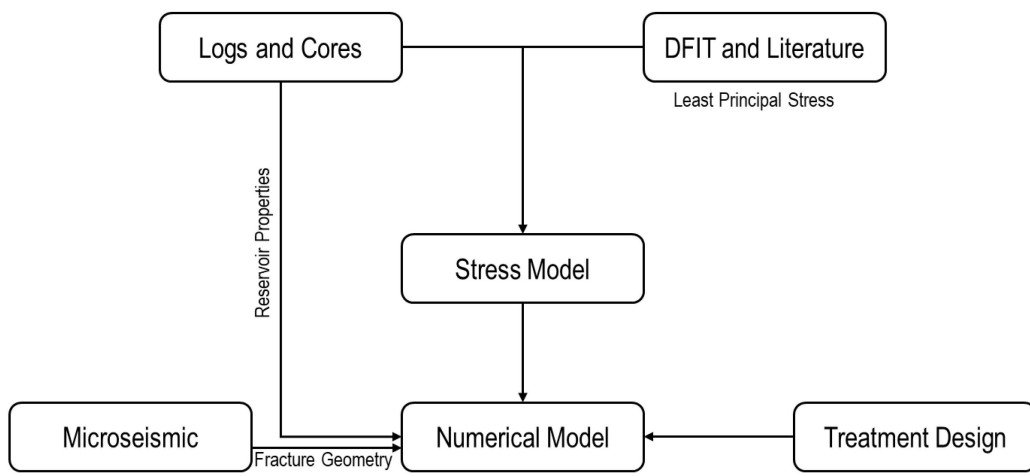

**Figure 2.** Schematic of the methodology and workflow used for the representative numerical model.

## 4.1. DFIT Analysis

The diagnostic fracture injection test (DFIT) has been used widely in unconventional reservoirs to assess fracture and reservoir properties prior to HF operation. Figure 3 displays a typical pressure profile of a DFIT with two distinct periods of before and after closure (BC and AC). The standard pressure curve recorded during the injection of fluids to the formation and pressure falloff data is analyzed to characterize the fractures and reservoirs' properties. The fracture closure event separated these two periods, and the fracture closure stress (least principal stress) and pore pressure can be inferred from the closure and post-closure pressure data. Generally speaking, minimum stress is horizontal in relaxed tectonic formations. The magnitude of the minimum principal stress, $S_{hmin}$, is estimated by identifying the 'contact pressure'—the pressure at which the walls of the fracture begin to contact as the fracture closes. In DFIT, vertical hydraulic fractures are often created perpendicular to the minimum stress due to the least-resistance path for hydraulic fractures to open. Economides and Nolte [36] state that for the ideal case (i.e., homogeneous formation), fracture closure pressure is the same as the minimum in situ horizontal stress (MHS). This case might not present the right condition in the fields (i.e., nonhomogeneous formation). The reason is that the walls of hydraulic fractures have roughness and do not mate perfectly as the walls come back into contact. As a result, the aperture of the crack never completely reaches zero. Instead, as the walls contact with the asperities of the fracture, a 'contact stress' pushes back on the closure, causing a nonlinearly increasing fracture stiffness (decreasing fracture compliance) [16,17]. Furthermore, fracture-closure pressure refers to the pressure at which the fracture is grossly closed. Although fracture closure pressure from DFIT analysis may be slightly greater than the actual value of minimum in situ horizontal stress, the obtained results can still be considered approximately equal to the minimum in situ horizontal stress. Therefore, in this paper, all fracture closure pressure obtained from visual inspection and nonlinear regression on an equation of diagnostic techniques, such as G-function, square-root-of-time, and log-log plot analyses, are assumed to be equal to the minimum in situ horizontal stress.

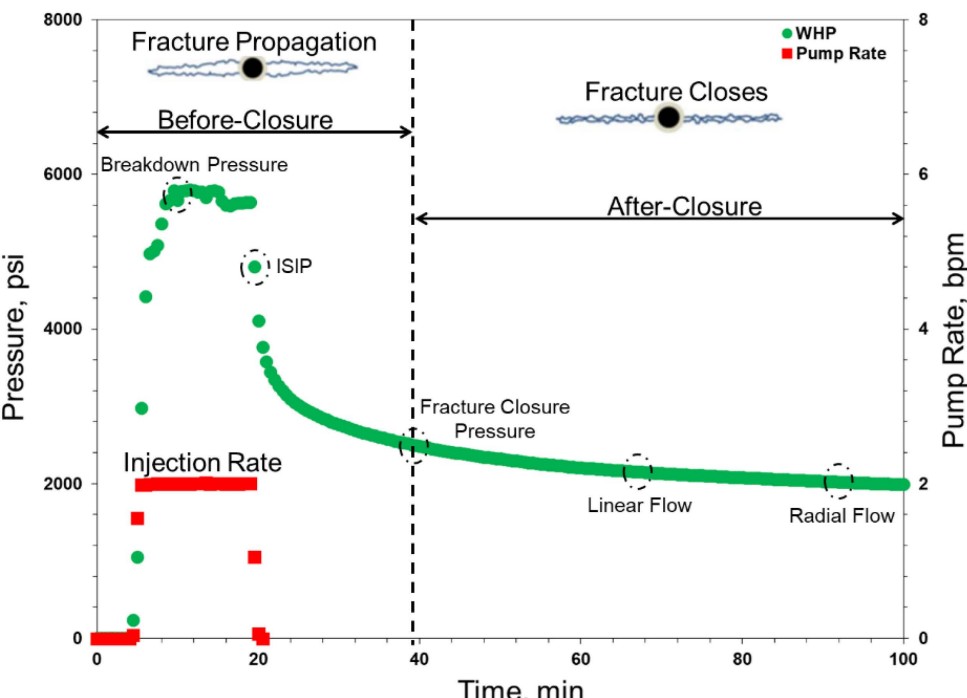

**Figure 3.** DFIT example of Meramec Formation includes a sequence of main events observed through pressure response before and after closure behavior (Ellafi and Jabbari, 2021).

The DFIT requires 4–5 days of shut-in to observe after closure (AC) and characterize the properties of the reservoir rock and propagated fractures. This is while a typical DFIT may take longer to reach radial flow; however, it was the operator's choice to run for this period. In contrast, a traditional pressure transient test, such as a buildup test, is impractical due to the long shut-in time required to reach the radial flow regime. An effective fracture treatment design can be tailored using an accurate stress model that was calibrated through DFIT outcomes. The results from a DFIT (least principal stress and pore pressure) are not limited to the application of HF but can be used in several ways, such as geologic carbon sequestration, nuclear waste repositories, and geothermal energy exploitation [37]. This section aims to apply DFIT methodology coupled with a geomechanical model to reduce uncertainty in DFIT outcomes, assist in monitoring pressure interference, and offset the well intervention.

In our case study, we used data from two unconventional wells from the Meramec and Woodford formations. These wells were completed using the plug and perf stimulation technique. DFITs (See Figure 4) were performed at the toe stage of Well #1 and #2. It is expected that the pressure rises linearly with the injection volume during the injection period, while the injection rate remains constant. The injection and pressure profiles for the DFITs were recorded at the surface and maintained at 2 bpm for 15 min in Well #1. For Well #2, it was 12 bpm for approximately 40 min, and the pressure falloff data were recorded during the next ten days for Well #1 and 4–5 days for Well #2.

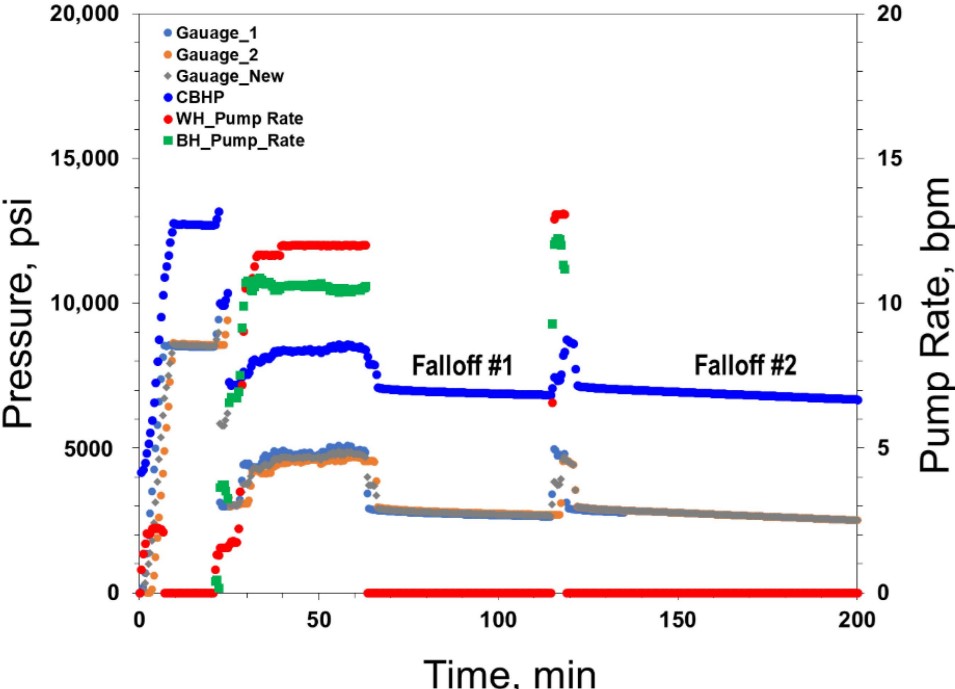

**Figure 4.** DFIT operations for Well #2, presenting injection and pressure profiles [37].

Barree et al. [33] and Liu and Ehlig-Economides [38] support a holistic methodology that may obtain lower closure pressure than the McClure et al. [16] approach that is built based on a numerical model incorporating variable fracture compliance. McClure et al. [16] assert that theoretical/modeling prediction builds confidence in analyzing DFIT data in low permeability formations, while the tangent method underestimates $S_{hmin}$ by a greater amount. The limitation of the tangent method compared with the compliance was highlighted in different field and experimental studies [18,39,40]. In our case study, both methods were used, and two MEMs were calibrated to estimate the hydraulic fracture height growth.

In Figure 5, The DFIT was analyzed using the tangent method, resulting in a pressure estimate of 6500 psi (first separation of the tangent). This will be used to calibrate the MEM. Figure 6 illustrates the analysis of the DFIT using the compliance method. The estimated stress is around 6830 psi from the graph; subtracting 75 psi resulted in 6755 psi, as recommended by McClure et al. [17]. Note that the difference between the two approaches is ~255 psi. Figure 7 illustrates the interpretation of pore pressure in a nearby well, 1200 ft distanced apart from Well #2, with an estimate of 5046 psi. The reason for estimating pore pressure from a nearby well is the short period of nearly five days for closure in Well #2, whereas the nearby well had a closure period of 10 days.

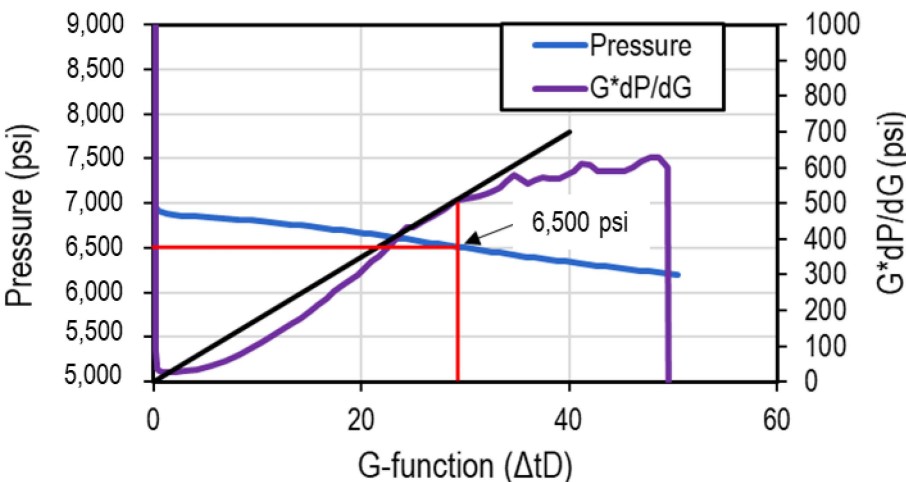

**Figure 5.** DFIT analysis using tangent method for Well #2 (Meramec Formation).

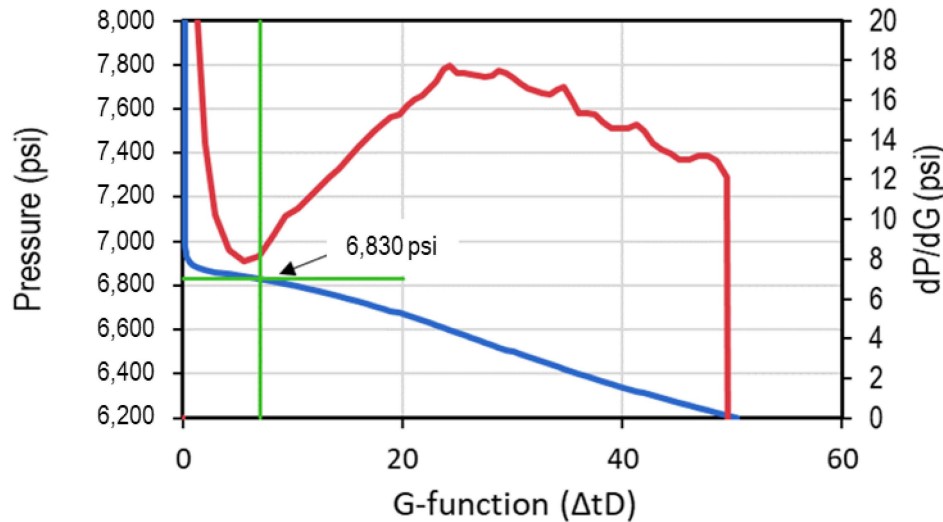

**Figure 6.** DFIT analysis using compliance method for Well #2 (Meramec Formation).

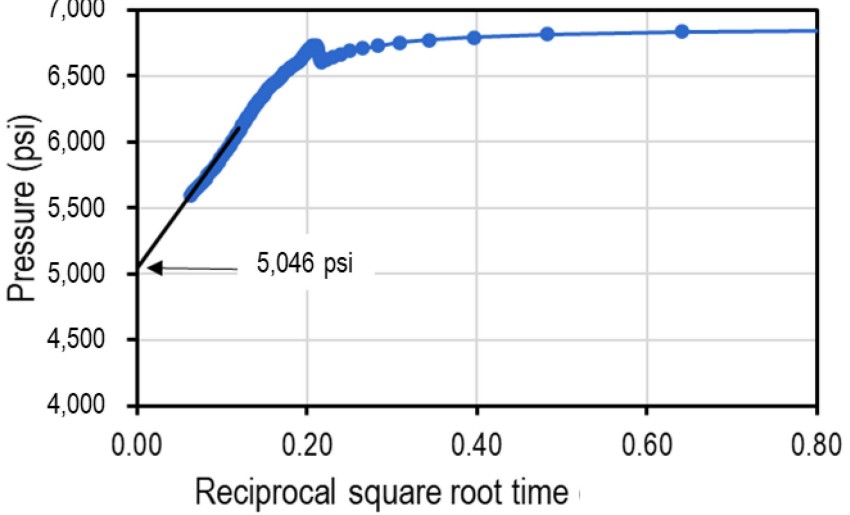

**Figure 7.** Post-closure reciprocal square root time for estimating pore pressure from a nearby well in the Meramec.

Figure 8 illustrates the pressure response after 10 days of pumping the DFIT in Well #1. Figure 9 is a zoomed-in pressure response to early time after the pump was shut-in in Well #1. The ISIP is estimated to be about 8578 psi. In Figure 10, the DFIT was analyzed using the tangent method, and the estimated stress was around 5950 psi. Figure 11 illustrates an attempt to interpret the DFIT using the compliance method; however, the trend is monotonically decreasing. According to McClure et al. [17], the closure event cannot be picked from a monotonically decreasing dP/dG curve. This can be due to two consecutive DFITs where the second one is smaller than the first one or the high leak-off due to the intersection of a highly conductive natural fractures system. Nadimi et al. [34] analyzed the DFIT response in a naturally fractured medium where a fault is intersected. Using numerical simulation, they found that the first separation from dP/dG represents the least principal stress. Following these conclusions and findings, the DFIT was interpreted. This approach is similar to the closure pick used by Pandey et al. [14].

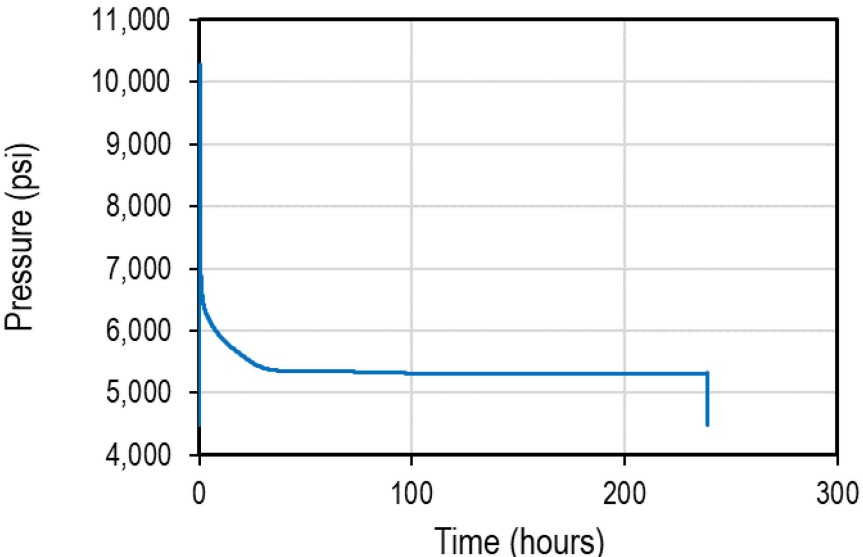

**Figure 8.** DFIT pressure changes during the 10 days for Well #1 (Woodford Formation).

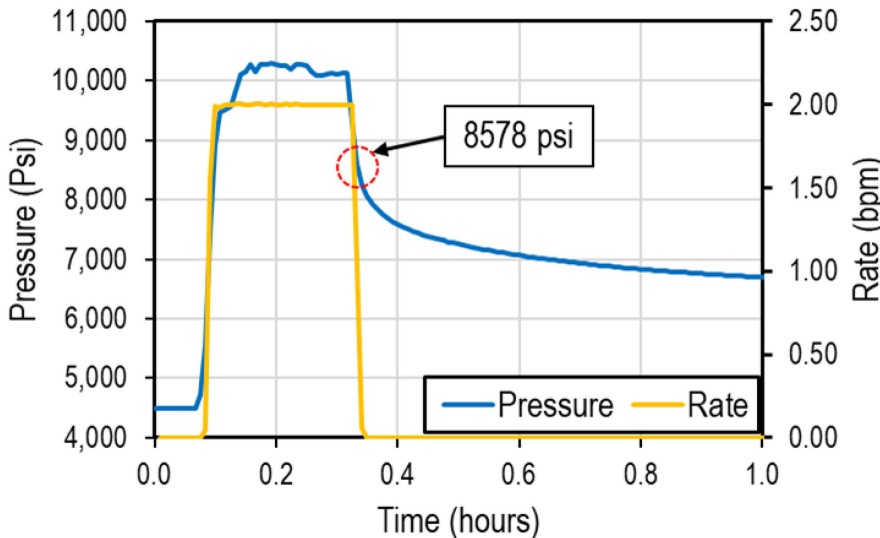

**Figure 9.** DFIT pressure evolution during injection and after shut-in for Well #1 (Woodford Formation).

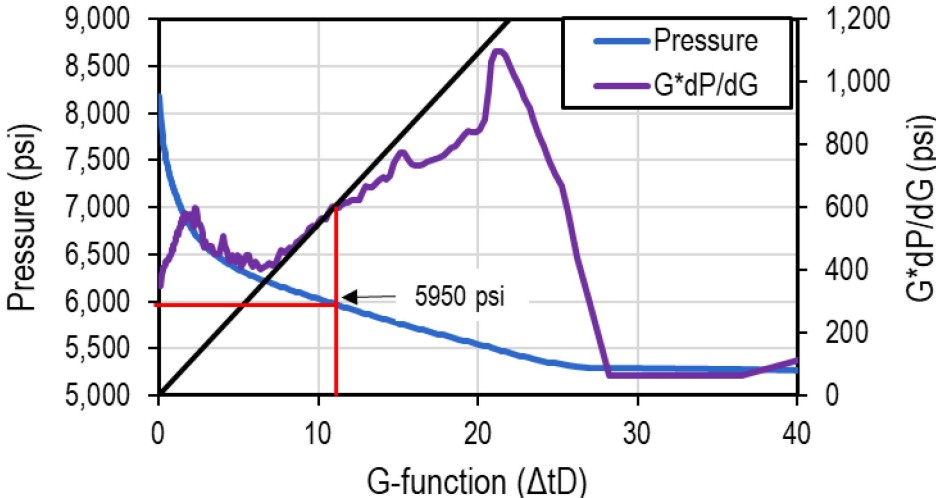

**Figure 10.** DFIT analysis using tangent method Well #1 (Woodford Formation).

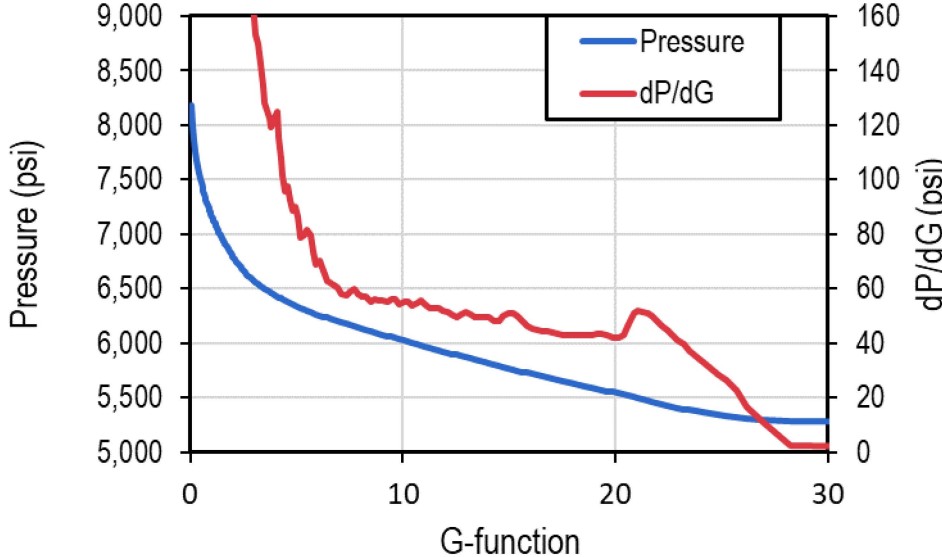

**Figure 11.** DFIT analysis using the compliance method for Well #1 (Woodford Formation).

Figure 12 illustrates the stress estimated in Well #1 with a value of 7600 psi that was confirmed with the root square approach. The existence of highly conductive faults and natural fractures has been mapped using microseismic data by Zoback and Kohli [41]. The pore pressure was estimated to be 5230 psi from the reciprocal time square plot in Figure 13.

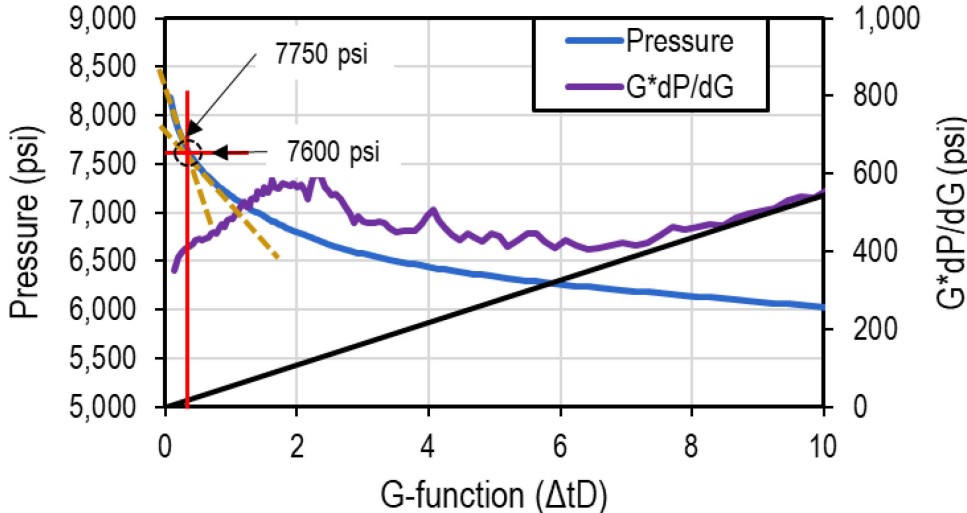

**Figure 12.** DFIT analysis and G-function plots for Well #1 (Woodford Formation).

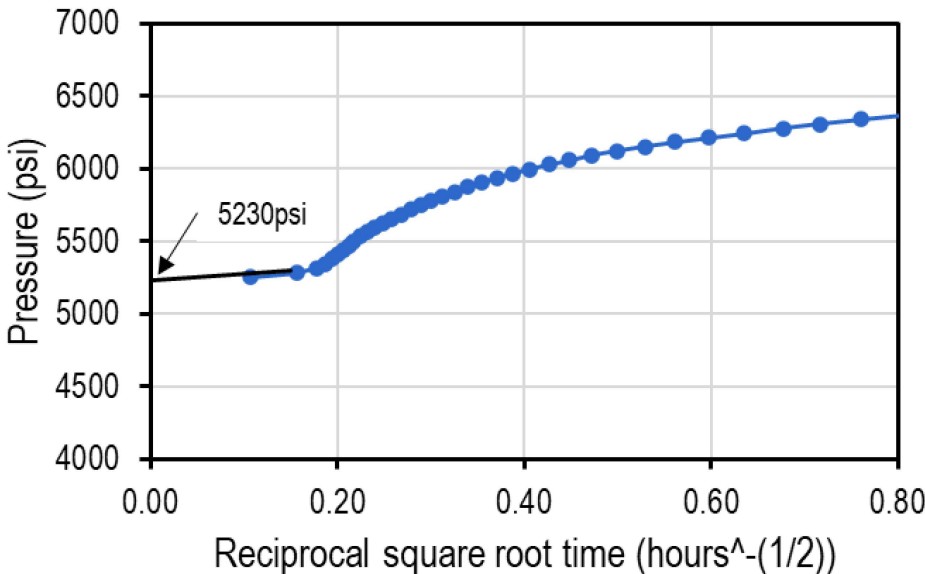

**Figure 13.** Post-closure reciprocal square root time for estimating pore pressure Well #1.

### 4.2. Stress Model

The stress profile was built by using the TI model Equation (5) and was compared with the results of the isotropic model Equation (2) and the frictional limit theory Equation (6), as well as the DFIT data and fracture gradient values from the literature. Note that only the least principal stress was calculated. The maximum horizontal stress was not calculated in this study.

The sonic and density log data (see Figure 14) were used to estimate the dynamic Young's modulus and Poisson's ratio using the isotropic assumption as follows [42]:

$$E_d = \frac{\rho V_s^2 \left(3V_p^2 - 4V_s^2\right)}{\left(V_p^2 - V_s^2\right)} \tag{7}$$

$$v_d = \frac{\left(V_p^2 - 2V_s^2\right)}{2\left(V_p^2 - V_s^2\right)} \tag{8}$$

where: $E_d$ Dynamic Young's modulus (psi), $v_d$ Dynamic Poisson's ratio (psi), $\rho$ bulk density from logs (lb/ft$^3$), $V_p$ compressional wave velocity from sonic logs (ft/μs), $V_s$ shear wave velocity from sonic logs (ft/μs).

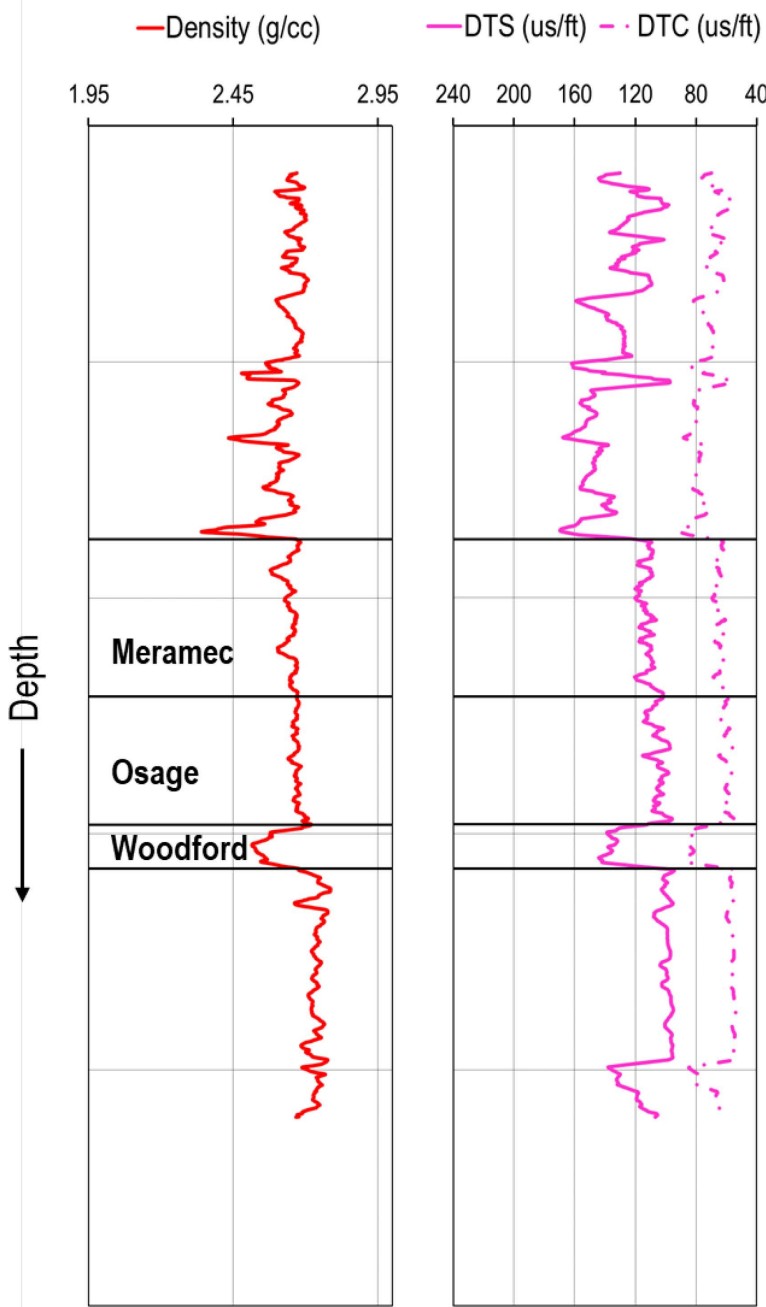

**Figure 14.** From left to right, density log, compressional slowness, and shear slowness variation with depth.

The logs from a vertical well, 1500 ft apart from Wells #1 and 2, were used for stress calculations. The density and compressional and shear sonic logs of this well are shown in Figure 14.

The static elastic properties are then estimated from correlations between the sonic measurements and core laboratory test results (see Figure 15). This data only belongs to the Meramec and Osage Formations. Such data was extracted by coring Well #1 and running triaxial testing. No coring was performed in Woodford Play. Thus, data from the literature was used.

As no cross-dipole sonic log was acquired in these wells, there were no fast and slow shear logs to calculate the dynamic stiffness properties. Thus, the isotropic model was used for both Meramec and the Osage formation, while the anisotropic elastic properties in the Woodford were estimated from the anisotropy ratio using core data reported by Sierra et al. [43] and Abousleiman et al. [44]. These references reported 10 data points from the Anadarko Basin with similar anisotropy ratios, which gives more confidence in using them. The vertical static Poisson's ratio was adjusted in zones where the well was cored and kept the same as the vertical dynamic Poisson's ratio in areas where data was unavailable. However, the horizontal Poisson's ratio was estimated from the anisotropy ratio (see Figure 16). The vertical and Horizontal Young's modulus was assumed to be the same in the Osage and Meramec Formations. In contrast, the anisotropy was estimated using correlation from literature core data reported in the same basin for Woodford Play.

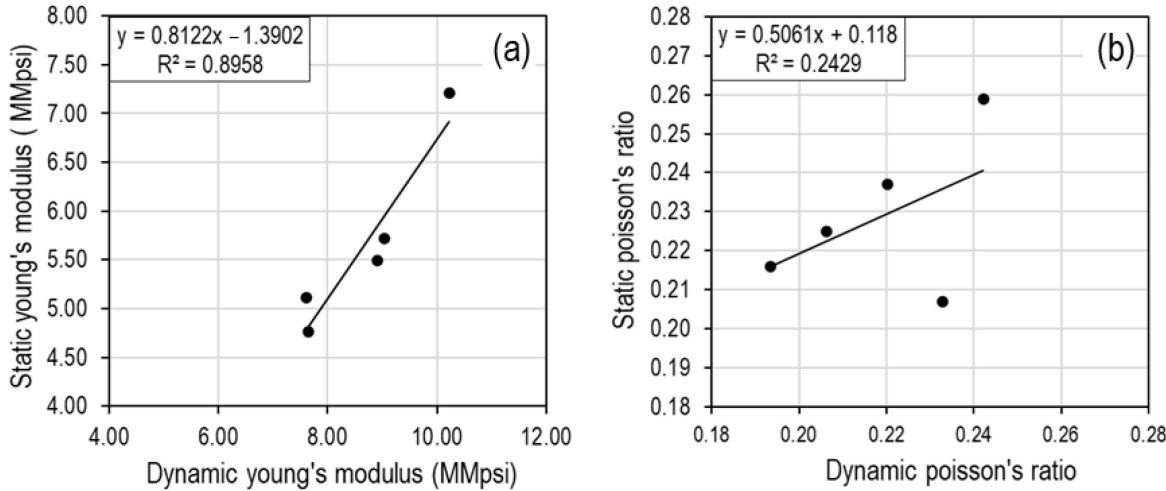

**Figure 15.** Relationship between dynamic and static (**a**) Young's moduli and (**b**) Poisson's ratio.

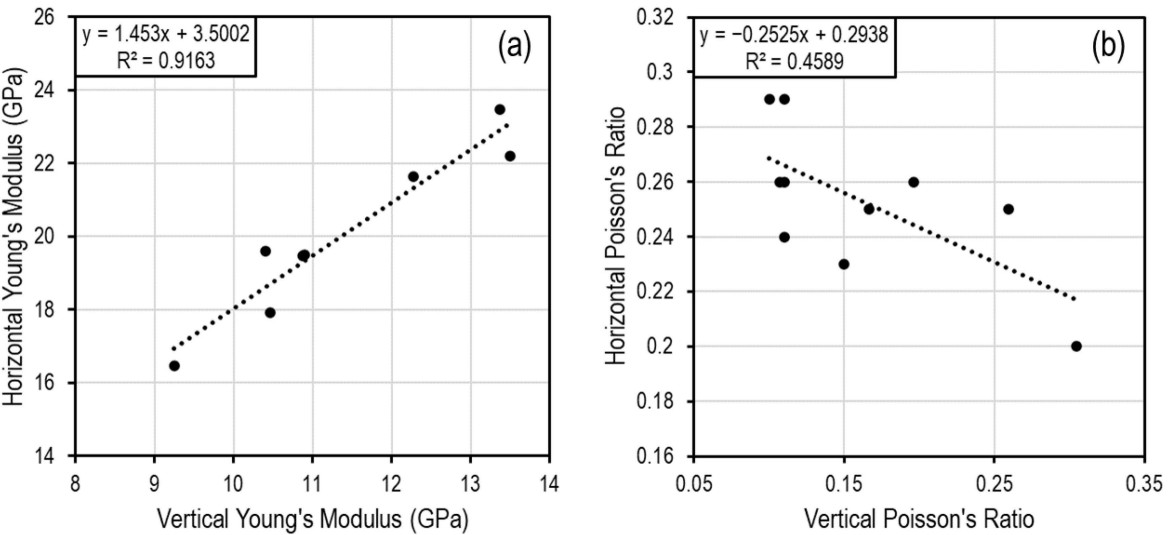

**Figure 16.** (**a**) Dynamic young's modulus and (**b**) Poisson's ratio anisotropy [43,44].

Equations (1), (2), and (4) were then applied to calculate the vertical and the minimum horizontal stresses, respectively.

The minimum possible values for stress were calculated from the frictional limit theory using a stress gradient of 1.1 psi/ft, pore pressure from DFIT, and a friction coefficient of 0.6 [45]. The value of 1.1 psi/ft was used because the well is located in a strike-slip faulting regime [32]. The use of 1.1 psi/ft is conservative for stress estimation.

The calculated stress profile assuming the isotropic model (yellow curve in the right track of Figure 17) shows very low values compared with the frictional limit theory. Even though the stress estimation from the tangent method aligns with the stress profile, these values are unlikely to be observed in the real field as it will result in the slip of critically stressed faults, leading to stress stabilization. For Woodford Formation, the stress values estimated from the frictional limit should be higher because the shale friction factor is expected to be lower than 0.6 [31]. Smith and Montgomery [46] reported a statistical analysis of measured stress contrast between sand and shale stresses. They noted that shales exhibit a higher stress gradient than sand, with 0.12 psi/ft on average and a mean of 0.105 psi/ft. This aligns with the anisotropic stress model.

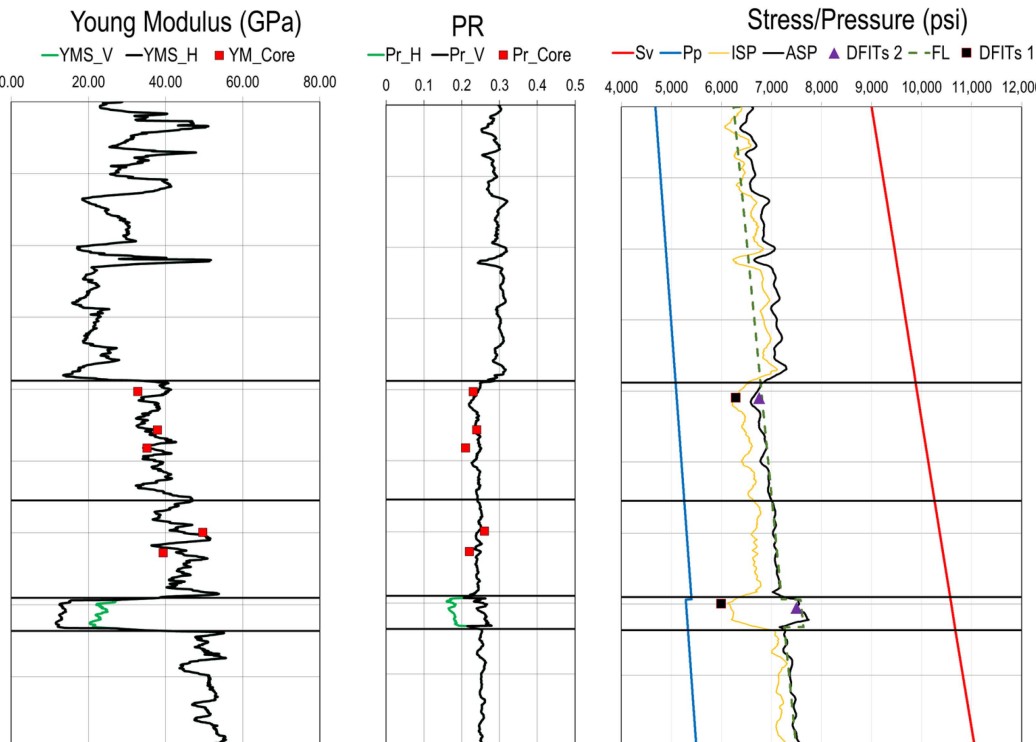

**Figure 17.** From left to right: calibrated static Young's modulus (horizontal and vertical), calibrated static Poisson's ratio, isotropic and anisotropic stress profiles with frictional limit stress estimation.

### 4.3. Numerical Modeling

The numerical modeling in this study was conducted using a fully 3D physics-based planar HF simulator that combines the reservoir and geomechanics simulator. The model was built in this simulator to account for the stress shadow between the two wells and for further studies, including stress changes and fracture design optimization.

The numerical model is based on the Linear Elastic Fracture Mechanics (LEFM) theory with fracture toughness as propagation criteria, meaning that an element is created when the stress intensity factor reaches or surpasses the fracture toughness [47,48]. The height growth is subject to stress contrast between layers. The code for the hydraulic fracture propagation is implemented using a finite difference numerical scheme while the reservoir code is implemented on a finite volume method. The simulator was validated against the analytical solutions of McClure et al. [49–51].

One stage per well was simulated for simplicity. The fracture geometry is tuned according to the bulk microseismic geometry using fracture toughness, relative fracture toughness, and Pressure Dependent Permeability (PDP) curves. The Instantaneous Shut-In Pressure (ISIP) was matched by adjusting the near-wellbore pressure drop. The stress profile was kept the same. The wellhead treating pressure (WHTP) match was ignored to avoid overfitting because the WHTP changes can be attributed to different reasons. The tuning process follows the steps and recommendations suggested by McClure et al. (2020;2021).

The inputs of the models are the following:

- Fracture toughness of 2000 $psi.in^{\frac{1}{2}}$ and relative fracture toughness of 0.5 $ft^{-0.5}$.
- Pore pressure and stresses from the MEM.
- Leak-off calculated from fluid flow using permeability (see Appendix A).
- Crossed relative permeability curves.
- A fracture mesh size of 100 ft and a wellbore mesh size of 15 ft.
- A geomodel with 1530 ft height, 10,000 ft length, and 1100 ft wide.
- A total of 122,518 grids (25i, 100j, 49k).

Note that the fracture propagation algorithm in the simulator separates the fracture propagation mesh size from the model mesh size. Even if the fracture mesh size is larger than the layer size, the propagation is solved accurately. This is achieved through a tracking algorithm on the fracture tip. The algorithm prevents a new element from forming as long as it is partially filled with fluids [52].

The Microseismic data allows relatively reliable mapping of induced fracture propagation from wells #1 and #2 in the Woodford and Meramec Formation. The fracture geometry summary obtained from microseismic interpretations is reported in Table 3.

**Table 3.** Interpreted bulk geometry from microseismic data.

| Well | | Half Length (ft) | | Height Growth (ft) | |
|---|---|---|---|---|---|
| | | East | West | Upward | Downward |
| #1 | Minimum | 1154 | 2030 | 547 | 53 |
| | Average | 1900 | 1620 | 644 | 157 |
| | Maximum | 2554 | 2920 | 744 | 315 |
| #2 | Minimum | 743 | 1814 | 76 | 345 |
| | Average | 1282 | 2173 | 154 | 483 |
| | Maximum | 1982 | 2664 | 214 | 614 |

The microseismic events were recorded on the heel section of the well only. The values in Table 3 summarize 11 stages from Well #1 and 13 stages from Well #2. It can be noted that fractures from Well #1 propagated upward from the Woodford to the Meramec with very low downward propagation. This is expected as the stress decreases in shallower depths. The fractures in Well #2 propagated downward. The downward propagation supports the existence of a stress barrier above the Meramec as little upward propagation was noted. The existence of a stress barrier agrees with the findings of Haustveit et al. [53] and Roberts et al. [54]. In their work, they reported the existence of a stress barrier of 1500 psi above the Meramec Formation. The lateral asymmetric growth is attributed to the existence of a depleted Well 2500 ft apart from Well #1. The hydraulic fracture growth of Well #1 aligns with the Fisher and Warpinski [12] data for the upward fracture growth in Blain County. Figure 18 shows the interpreted microseismic events, and the upper figure shows the fracture height from Well #1. The lower figure shows the fracture height from Well #2.

Figures 19 and 20 illustrate a typical treatment stage for Well #1 and #2, respectively. The pressure was recorded at the surface with a reported ISIP of 4066 psi and 3162 psi for Well #1 and #2, respectively. Note that these measurements are from the first stages next to the toe of the well, where the stress shadow is minimal.

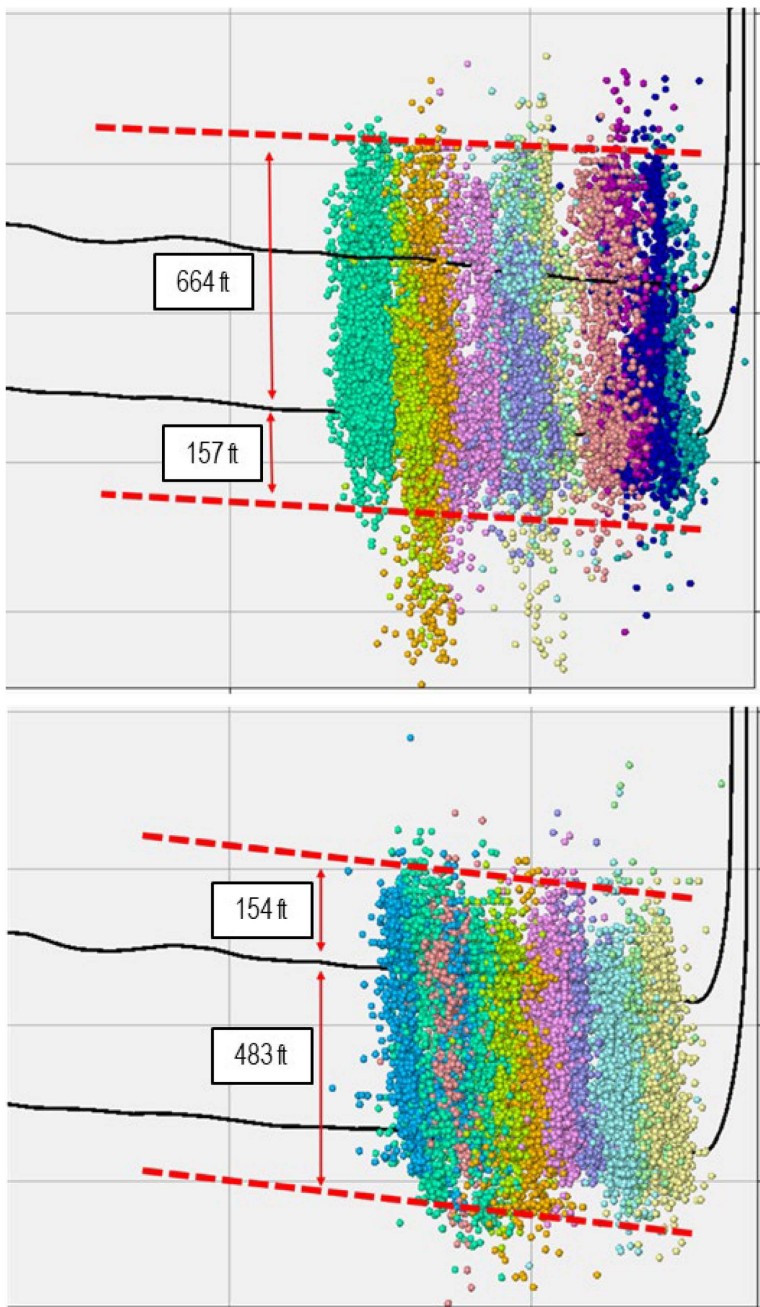

**Figure 18.** Microseismic event for height estimation. The upper figure shows the height from Well #1 and the lower figure shows the height from Well #2.

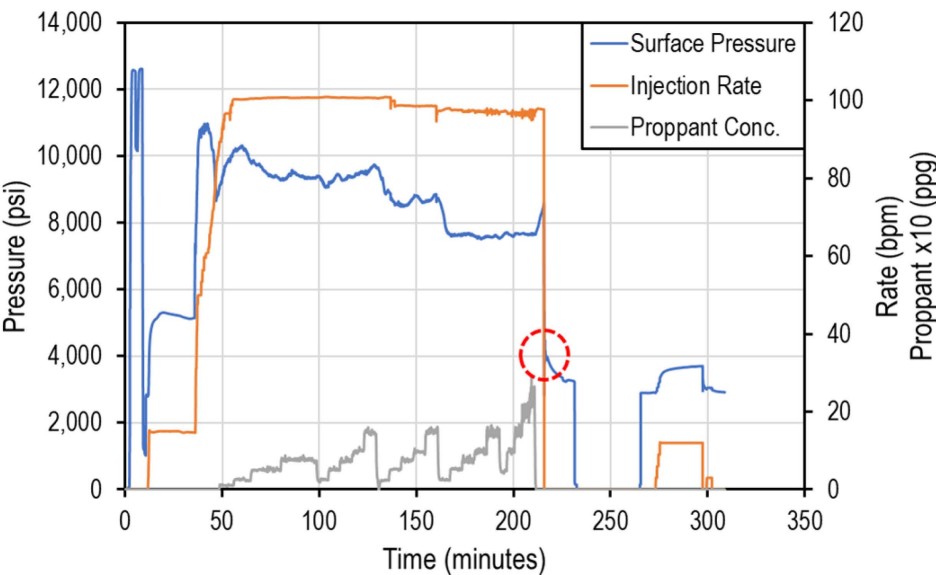

**Figure 19.** Typical treatment in Well #1.

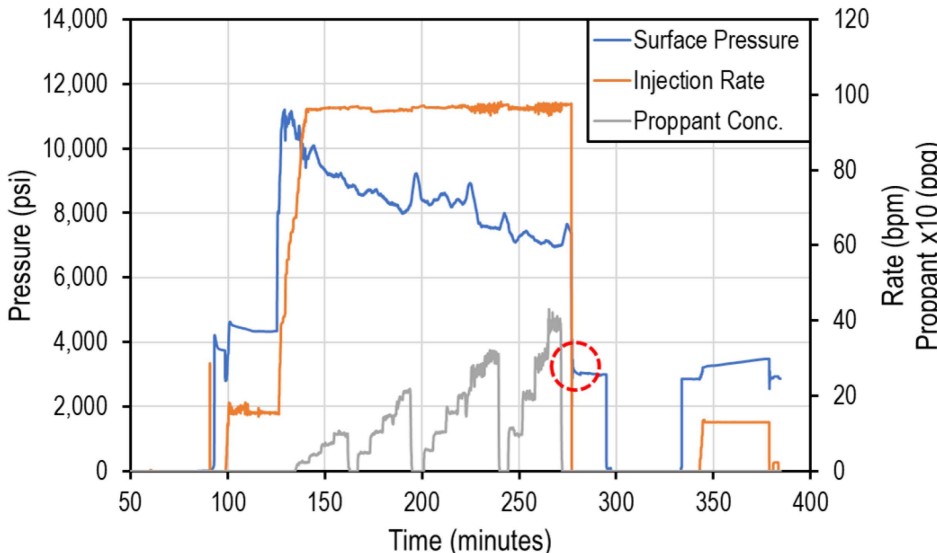

**Figure 20.** Typical treatment in Well #2.

## 5. Results and Discussion

The stress profile generated using the anisotropic stress model aligns with findings from different researchers [37,45,55,56]. The values also support the upward and downward growth of fractures seen from microseismic data, the ISIP values from the HF treatment in early stages, and reported data by others [4,41,45,57,58]. Lund Snee & Zoback [59] presented a comprehensive study on the stress state in North America. They proposed a diagram for the principal stress ratio by vertical stress as a function of pore pressure and the relative stress magnitude coefficient. They noted that with the assumption of hydrostatic pore pressure, the minimum horizontal-to-vertical stress ratio is larger than 0.5 in a strike-slip regime and larger than 0.6 for the Anadarko Basin, which is located at a relative stress magnitude of approximately 1.5 from the stress map. They also noted that this ratio is higher in clay-rich environments due to stress relaxation. These findings support the use of the anisotropic mechanical earth model.

Ma and Zoback [45] analyzed ISIP values from different stages (as a proxy for least principal stress values) of the same well in the Woodford Formation. Their analysis reports a wide range of ISIP values in these stages. The ISIP values were correlated with lithology variation along the well trajectory, which means that if the clay and kerogen content increase, the ISIP values will increase (See Figure 21). This suggests that different stress values may be encountered along the horizontal section of the well. Note that along the lateral section, the ISIP values are always higher than the frictional limit, which aligns with our findings.

Ma and Zoback's [45] results were reproduced for this case study. Figure 22 represents the variation of ISIP and gamma-ray along the horizontal section. The graph shows some correlation between the ISIP values and the gamma-ray. The average value of ISIP along the lateral is 4717 psi and a standard deviation of 1214 psi, a minimum value of 3391 psi, a maximum value of 9803 psi, and a median of 4429 psi (Note that these values were recorded at the surface). Even though the correlation is not consistent along the entire horizontal section, the effect of different clay volumes on the stress cannot be ignored [41].

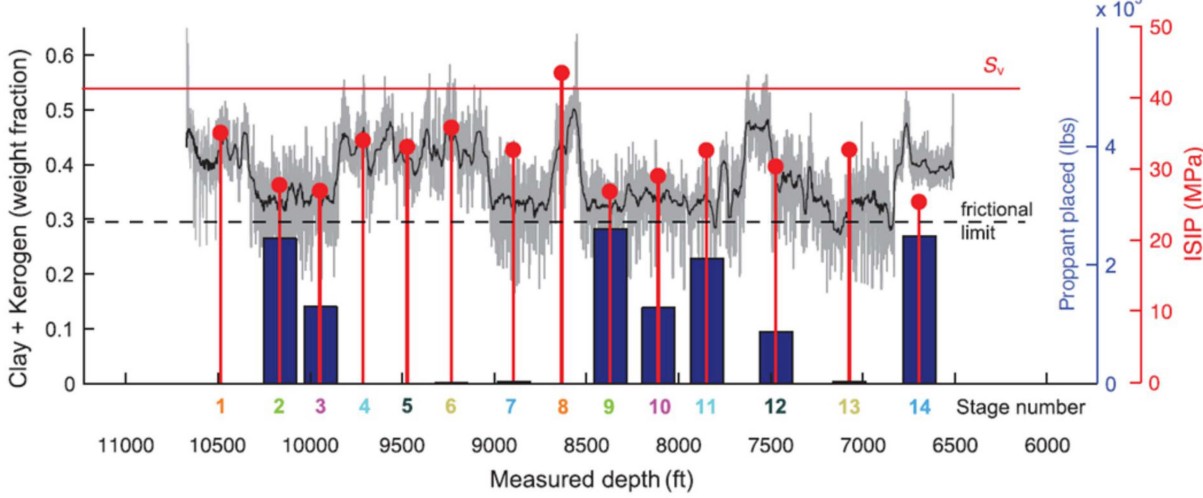

**Figure 21.** Variation of clay and kerogen content, ISIP, and Proppant placed along the horizontal section of one well [45].

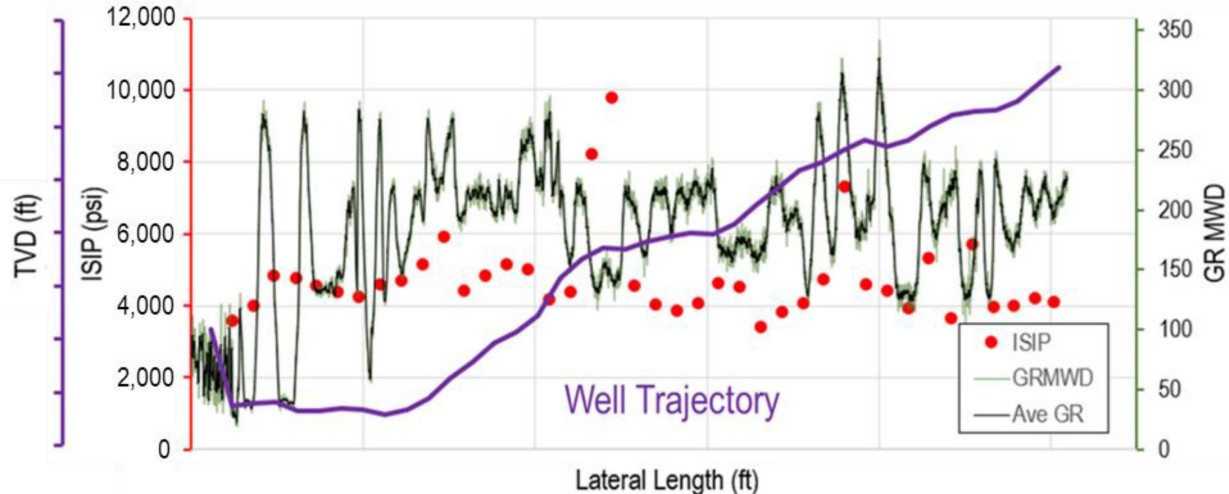

**Figure 22.** Gamma-ray measured while drilling and ISIP values variation along the lateral in Well #1, the purple line is the well trajectory.

The hydraulic fracture propagation in brittle materials does not require high net pressure values. A net pressure of 200–300 psi can be enough to drive the fracture to propagate further [31,60]. Since the recorded ISIP of the DFIT job is 8578 psi, the stress values should not be very low compared with this value. ISIP can be calculated as follows [36]:

$$ISIP = Net\ pressure + Minimum\ Stress + Near\ wellbore\ Complexity \tag{9}$$

The DFIT was conducted using 2 bbl/min with a total volume of 30 bbl, which reduces the near-wellbore complexity effect. The net pressure can be influenced by several factors, such as near-wellbore complexity and perforation pressure drop. Thus, an exact estimate for net pressure cannot be obtained.

The net pressure and geometry can be calculated using the Perkins and Kern-Nordgren (PKN) model as follows [61–63]:

$$x_f = 0.68 \times 16^{-\frac{1}{5}} \left( \frac{q^3 E'}{\mu h_f{}^4} \right)^{\frac{1}{5}} t^{\frac{4}{5}} \tag{10}$$

$$w_0 = 2.5 \times 2^{-\frac{1}{5}} \left( \frac{q^2 \mu}{E' h_f} \right)^{\frac{1}{5}} t^{\frac{1}{5}} \tag{11}$$

$$P_{net} = 2.5 \times 64^{-\frac{1}{5}} \left( \frac{E'^4 q^2 \mu}{h_f{}^6} \right)^{\frac{1}{5}} t^{\frac{1}{5}} \tag{12}$$

where: $q$ pumping rate (m$^3$/s), $\mu$ viscosity (Pa.s), $t$ injection time (s), $w_0$ fracture width (cm), $P_{net}$ net pressure (Pa).

The DFIT job was pumped with a rate of 2 bbl/min and a viscosity of 1 cp (0.001 Pa.s). The calculated net pressure and fracture geometry (fracture half-length and width) for different Young's moduli and Poisson's ratios are illustrated in Figure 23. The fracture is expected to have a relatively close height and half-length because the fracture is not contained at a small, injected volume of 30 bbl. Thus, the net pressure varies between 100–250 psi. With ISIP values of 8578 psi, the minimum stress is expected to be around 8300 psi from equation 20. This is while the values from DFIT using the tangent method suggest a value of 5900 psi. Even though this value matches the stress profile from the isotropic model, it is below the frictional limit and results in a higher net pressure of around 2660 psi. These results contradict the predictions from the PKN model and the literature [45,56].

Stress variation with clay content and the expected results from the PKN model suggest that the interpretation of the DFIT using the tangent method can be misleading and cannot be relied upon to calibrate the stress profile. This value approximates the PKN model, the ISIP, and data from the literature [55]. The values in the literature report that the stress gradient at the Woodford Formation of the Anadarko Basin is around 0.86 psi/ft.

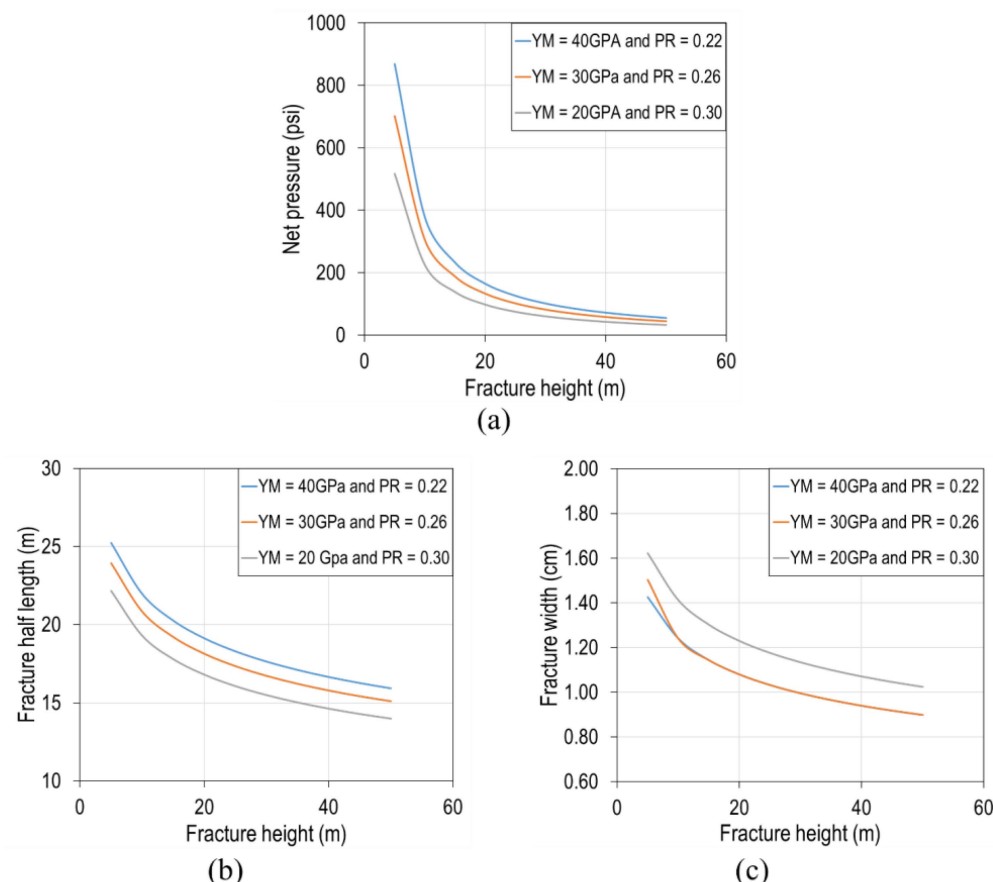

**Figure 23.** Calculated (**a**) net pressure, (**b**) fracture half-length, and (**c**) fracture width for different fracture heights, Young's modulus, and Poisson's ratio values.

Figures 24 and 25 illustrate the predicted fracture geometry using the numerical simulator. Figure 24 represents the geometry of the anisotropic stress model, whereas Figure 25 illustrates the geometry of an isotropic stress model. In Figure 24, the predicted fracture geometry has a 665 ft height for Well #1 with an upward growth of 630 ft, a downward growth of 35 ft, and a lateral length of 2648 ft. Well #2 shows a total height of 581 ft with an upward growth of 162 ft and a downward growth of 419 ft and lateral growth was 2168 ft. Figure 25 reports the predicted fracture geometry for the isotropic stress model. Well #1 had a total height growth of ~500 ft with an upward growth of 392 ft, a downward growth of 107 ft, and a lateral growth of 6919 ft. For Well #2, the total predicted height was 486 ft with an upward growth of 219 ft, a downward growth of 267 ft, and a lateral growth of 4467 ft.

The fracture length and height are different for the two models. This is a result of fracture containment. In the isotropic model, the stress is low in the Woodford formation. This results in containment of the fracture height within the thin formation and results in longer fractures compared with the anisotropic case. In the anisotropic case, the fracture propagates upward from Woodford and downward from the Meramec because the stress is high in Woodford and in the formation above the Meramec. This results in longer fracture height and shorter fractures compared with the isotropic stress model case.

The fracture geometry predicted from the anisotropic model is in close proximity to the fracture geometry estimated from microseismic events. However, the fractures from the isotropic stress profile had longer laterals with shorter heights. With the arguments presented with respect to the analysis of the microseismic, frictional limit, stress map, net pressure estimation, and stress variation with lithology, it can be said that the anisotropic stress model, along with DFIT analysis from the compliance method, aligns with the recorded microseismic and are more appropriate to use in similar case studies.

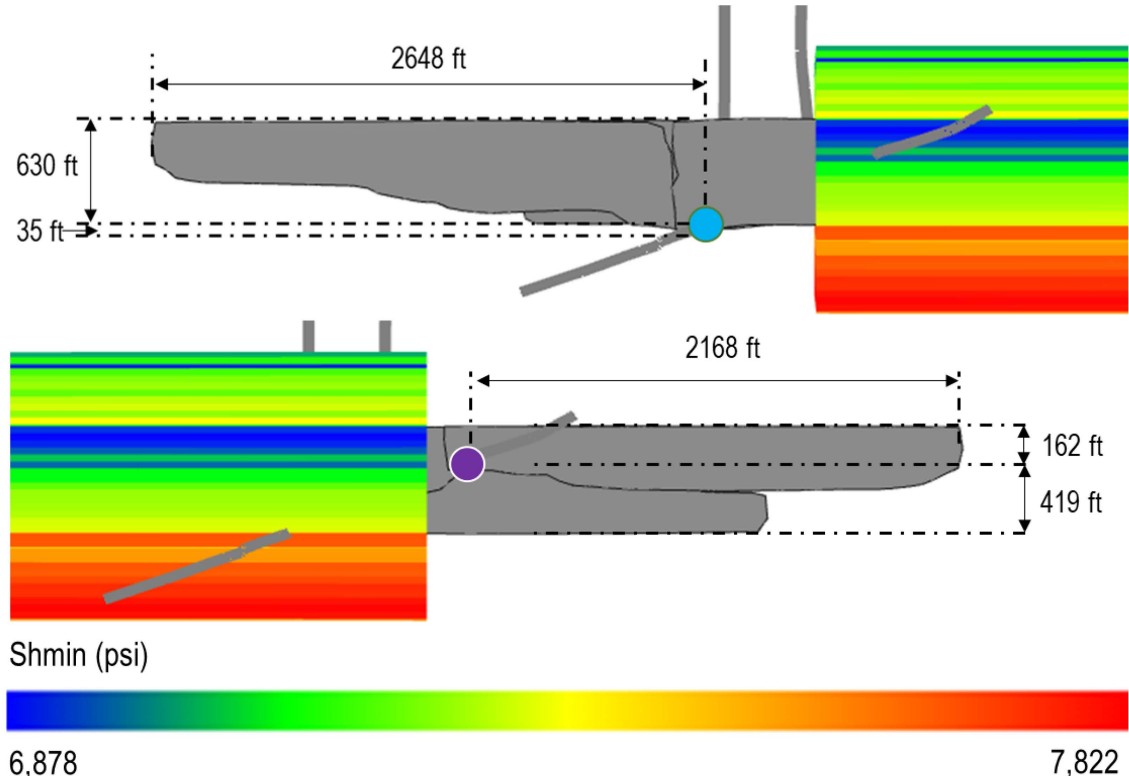

**Figure 24.** Fracture geometry predicted from numerical simulations for the anisotropic stress profile, in bleu Well #1, in purple Well #2.

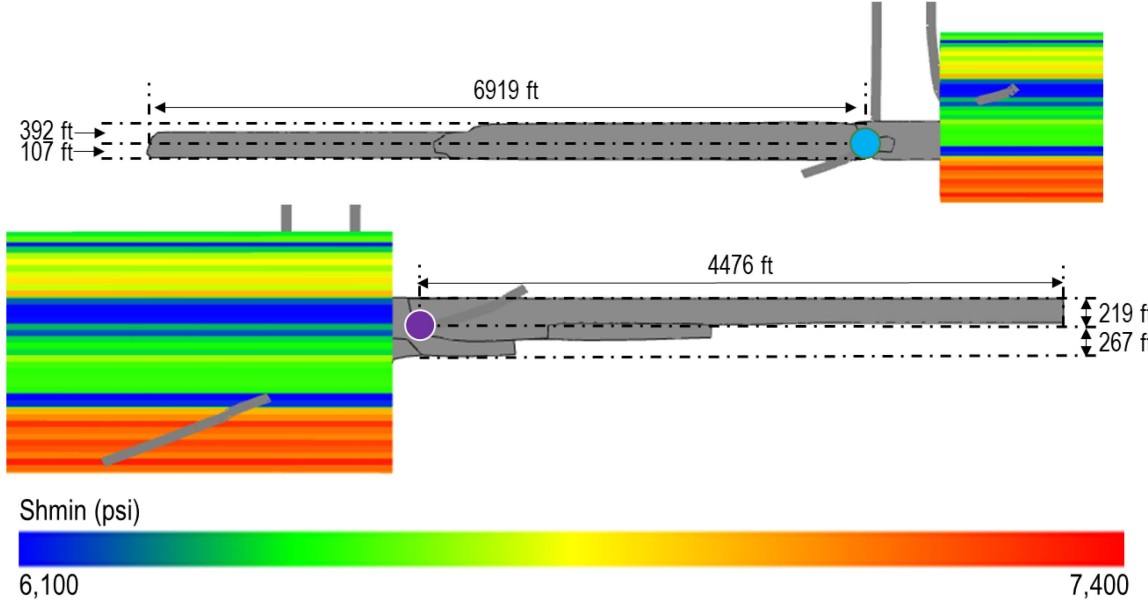

**Figure 25.** Fracture geometry predicted from numerical simulations for the isotropic stress profile, in bleu Well #1, in purple Well #2.

## 6. Conclusions

This work presented a case study in the Anadarko Basin of Oklahoma, where fracture height was estimated using different stress estimation models and DFIT interpretations and compared with fracture height from microseismic events. The following conclusions were drawn from this study:

- The method of DFIT analysis may lead to erroneous estimation of stress because of the geological complexity of the formation, so it is essential to QC that the DFIT results use values that are not very low compared with the ISIP, World Stress map, values from the literature, and the frictional limit theory.
- Several interpretation techniques for the DFIT need to be conducted to reduce the uncertainty of one method over the other.
- In heterogeneous formations with different clay and kerogen content, it is preferable to conduct several DFITs and account for kerogen and clay content to ensure reliable estimates.
- The anisotropic model can capture the higher stress values in formations with high clay content, while using the isotropic approach can lead to very low values.

This study can be extended into a calibrated reservoir model to assess different scenarios of well spacing.

**Author Contributions:** Each author has contributed to the present paper. A.M. was responsible for preparing the methodology, analyzing the data, validating, writing the paper, and drawing the final conclusion. A.E. was responsible for analyzing and DFIT Interpretation. V.R. reviewed the paper and discussed the results and methodology. H.J. provided the data and discussed the results. All authors have read and agreed to the published version of the manuscript.

**Funding:** This research was possible through North Dakota Industrial Commission (NDIC) for the financial support contract NDIC G-045-89.

**Data Availability Statement:** The data presented in this study are available on request from the corresponding author. The data are not publicly available due to confidentiality by the operator.

**Acknowledgments:** The authors would like to acknowledge the North Dakota Industrial Commission (NDIC) and Petroleum Research Fund, for their financial support of this work through the contract NDIC G-045-89. We are also thankful to the Reservoir Engineering Management of the Continental Resources (CLR) for providing us with their engineering insights and data. We would like to acknowledge Mouna Benabid for her help in plotting the microseismic data. We also acknowledge Neal Nagel and Marisela Sanchez-Nagel for reviewing the paper.

**Conflicts of Interest:** The authors declare no conflict of interest.

## Appendix A

The used core data in this approach are reported in Table A1.

**Table A1.** Measure cores rock properties.

| Sample | Permeability | Porosity | Saturation | |
|---|---|---|---|---|
| **Depth** | **Millidarcies** | **%** | **%** | |
| feet | Air | Ambient | Water | Oil |
| 9744.97 | 0.000135 | 1.9 | 76.7 | 18.1 |
| 9751.99 | 0.000027 | 1.5 | 57.5 | 26.7 |
| 9763.96 | 0.000136 | 2.1 | 68.1 | 12.5 |
| 9780.08 | 0.000445 | 4.4 | 62.9 | 26.4 |
| 9820.00 | 0.000159 | 4.0 | 51.5 | 31.9 |
| 9825.97 | 0.000118 | 2.3 | 49.2 | 35.0 |
| 9839.99 | 0.000075 | 2.2 | 81.7 | 15.6 |
| 9860.03 | 0.000109 | 3.9 | 48.7 | 28.1 |
| 9880.08 | 0.000119 | 4.2 | 62.8 | 28.0 |
| 9886.01 | 0.000010 | 1.9 | 66.6 | 21.5 |
| 9899.00 | 0.000139 | 1.4 | 77.3 | 15.2 |
| 9904.97 | 0.000132 | 2.4 | 32.2 | 36.4 |
| 9923.03 | 0.000056 | 2.3 | 55.3 | 26.0 |
| 9945.98 | 0.000399 | 3.5 | 71.1 | 23.0 |
| 9955.00 | 0.000193 | 5.2 | 15.0 | 34.3 |
| 9999.01 | 0.000116 | 2.2 | 49.8 | 31.7 |
| 10,007.01 | 0.000380 | 1.2 | 36.8 | 14.4 |
| 10,014.04 | 0.000067 | 2.1 | 50.6 | 29.8 |
| 10,022.03 | 0.000141 | 1.6 | 62.2 | 22.5 |
| 10,034.04 | 0.000179 | 3.6 | 79.5 | 17.5 |
| 10,060.01 | 0.000208 | 1.3 | 74.6 | 6.8 |
| 10,110.06 | 0.000082 | 1.3 | 62.8 | 11.3 |
| 10,122.01 | 0.000085 | 1.8 | 78.2 | 6.2 |
| 10,133.94 | 0.000099 | 0.4 | | |
| 10,150.96 | 0.000066 | 1.3 | | |
| 10,163.96 | 0.000003 | 0.6 | | |
| 10,247.99 | 0.000131 | 2.3 | 85.7 | 11.8 |
| 10,286.95 | 0.000004 | 0.8 | | |

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
