# Peer review of "Anisortopic Modeling of Hydraulic Fractures Height Growth in the Anadarko Basin"

_2673-3161, doi:10.3390/applmech4010004_

Round 1

Reviewer 1 Report

The overall effort is admirable, considering the complexity of the subject. I would not question the applicability of the methods used in the analysis. However, I feel the results and conclusions do not provide more meaningful insight into the theoretical understanding of this subject or valuable guidance for practical hydraulic fracturing treatments, if they do not add more confusion. I am making a few comments for the authors to consider.

(1) Many factors influence fracture height in heterogeneous formations: stress, young modulus, toughness, leakoff, to name a few. Of course, one could argue stress may be the most dominant effect. The other effects should be somewhat addressed or delineated for the presented analysis to be valid. 

(2) The in-situ stress in a formation can be complex, as the authors may recognize in the paper. Various methods were presented in the past to construct the stress model. Using these models to construct the stress model for these two particular wells is understandable. This seems to be one of the main objectives of this paper. However, the data appear to be somewhat sketchy. The presentation could be organized better and presented more coherently. My primary concern is the selection of the model and conclusion. Isn't the DFIT still the most reliable direct measurement of the in-situ stress, despite some variations for various specific methods? I am somewhat confused about how the authors could rely on other theoretical models to correlate the DFIT stress. The presented simulation cannot be used to validate the measurement, considering the model itself is very crude.

(3) The leakoff coefficient is a critical parameter in simulating fracture geometry. It is well-known that one cannot use the result obtained from the lab-measured permeability, etc. In practical application, most would infer the data from the DFIT analysis, which is considered more reliable. Why is the DFIT data not used here since the information is also in the pressure decline curves?

Author Response

Thank you so much for your comments. I do appreciate the time you took to read the paper and review it.

(1) Many factors influence fracture height in heterogeneous formations: stress, young modulus, toughness, leakoff, to name a few. Of course, one could argue stress may be the most dominant effect. The other effects should be somewhat addressed or delineated for the presented analysis to be valid.

It is true that several factors influence fracture height in heterogeneous formation. The effect of these parameters is different from one case to another. In our study, we note the use of higher viscosity fluid (25cp), which can result in a larger length and lower leakoff, which makes the leakoff effect small. Through the microseismic data, it can be noted the repeatability of the fracture growth pattern. This implies very strong barrier that cannot exists except for a stress barrier. If the effect was toughness or Young Modulus. Stress shadow would have broken the barrier due to toughness or elastic properties. Note that the wells had 40 stages in total, which can accumulate higher stress shadow values. The average stress shadow build-up in the Meramec is 200psi per stage (Haustveit et al., 2022).

(2) The in-situ stress in a formation can be complex, as the authors may recognize in the paper. Various methods were presented in the past to construct the stress model. Using these models to construct the stress model for these two particular wells is understandable. This seems to be one of the main objectives of this paper. However, the data appear to be somewhat sketchy. The presentation could be organized better and presented more coherently. My primary concern is the selection of the model and conclusion. Isn't the DFIT still the most reliable direct measurement of the in-situ stress, despite some variations for various specific methods? I am somewhat confused about how the authors could rely on other theoretical models to correlate the DFIT stress. The presented simulation cannot be used to validate the measurement, considering the model itself is very crude.

Thank you for your comment. The data that we have plotted are the same data given to us by the operator. The interpretation of the data using the isotropic model did not allow us to re-produce the same fracture geometry as the fracture is going to be contained within the lower stress zone estimated by the tangent method approach. Looking at the literature, we find a consistency that the stress is higher in formation with higher clay content. The estimated stress in Woodford is consistently higher in the literature aswell. This pushes us to look into other approaches to interpreting the DFIT data. We tried the compliance method, which was not applicable due to the continuously monotonic dp/dG graph (Note that the method was applicable and clear in the Meramec). Thus, we used the first separation as a proxy for to end of linear flow when the fracture closes. The theoretical approaches were ways to constrain the approach used. They were used to check if our interpretation results and method are acceptable or not. For instance, The stress ratio between minimum stress (which is minimum horizontal) and the maximum stress (maximum horizontal) should not exceed a certain factor, or an earthquake will occur. This is based on the frictional limit theory, which has been proven in both the world stress map and the explanation of induced seismicity. It is true that DFIT is the most reliable direct measurement; however, The data need to be looked at from a theoretical point of view as well because a lot of things can happen in a DFIT like intersecting a fault or high permeability streak (like pre-existing fracture). This will make the DFIT non-interpretable. Thus we try to constrain our estimate using theoretical approaches and model keeping in mind the DFIT response.

(3) The leakoff coefficient is a critical parameter in simulating fracture geometry. It is well-known that one cannot use the result obtained from the lab-measured permeability, etc. In practical application, most would infer the data from the DFIT analysis, which is considered more reliable. Why is the DFIT data not used here since the information is also in the pressure decline curves?

The leakoff coefficient is indeed a critical parameter in simulating fracture geometry. The DFIT conducted in the Meramec was short and to be able to estimate permeability. Thus we cannot use DFIT to estimate permeability due to the short response. The permeability in the formation is very low to affect the fracture geometry as we are dealing with permeabilities in the level of mD. The permeability would not have an effect on the fracture height, which is the aim of this study.

Haustveit, K., Elliott, B., & Roberts, J. (2022, January 25). Empirical Meets Analytical-Novel Case Study Quantifies Fracture Stress Shadowing and Net Pressure Using Bottom Hole Pressure and Optical Fiber. Day 2 Wed, February 02, 2022. https://doi.org/10.2118/209128-MS

Reviewer 2 Report

This paper by Ahmed Merzoug et al. presented a case study in the Anadarko Basin of Oklahoma, where fracture height was estimated using different stress estimation models and DFIT interpretations and compared to fracture height from microseismic events. It is recommended to be published after the following issues are resolved.

1. Abstract section for the relevant principles of the description is a little too much, it can be appropriate to delete the details.

2. In lines 211, 212, 215, and 309, Figure 5.6.7 is bold and italic, suggesting a unified format for the whole text.

3. Regarding the correction parameters of 6500Psi in line 211, I hope to further elaborate on why this point is chosen.

4. From lines 227 to 243, there are as many as five pictures in the text description. The distance between the text and the corresponding picture is too far, so can the text be interspersed adequately between the pictures

5. When we introduced the model in Section 4.3, the curve on line 342 was placed here somewhat out of parameter Settings.

Author Response

Thank you so much for your comments. I do appreciate the time you took to read the paper and review it.

  1. Abstract section for the relevant principles of the description is a little too much, it can be appropriate to delete the details.

Thank you so much for the comment. The abstract is indeed a little too much. We want to show all the work done in the paper so the reader can have ab idea about the workflow and the information offered. The abstract was made for a general audience so they have an idea of the work that was done.

  1. In lines 211, 212, 215, and 309, Figure 5.6.7 is bold and italic, suggesting a unified format for the whole text.

Thank you so much for pointing this out. This was fixed

  1. Regarding the correction parameters of 6500Psi in line 211, I hope to further elaborate on why this point is chosen.

6500psi was chosen as the first separation from the tangent line. Which, according to the tangent method, represents the least principal stress.

  1. From lines 227 to 243, there are as many as five pictures in the text description. The distance between the text and the corresponding picture is too far, so can the text be interspersed adequately between the pictures

The Text was broken into two parts to solve this issue.

  1. When we introduced the model in Section 4.3, the curve on line 342 was placed here somewhat out of parameter Settings

The curves discussed are X-curved, meaning they go from the minimum of the endpoint to the maximum in both directions.

Reviewer 3 Report

Reviewer's comments are in a file attached.

Author Response

Thank you so much for the time and effort to review the manuscript, I really appreciate it.

  1. Introduction: Thank you so much for the suggested papers, I will make sure to add them to the paper as they make very good arguments for the prediction of productivity.

The problem statement was fixed

  1. Methodology: Thank you for recommending another paper. If an engineer needs values of Young’s Modulus. They can use both as a lot of recent simulators can account for elastic anisotropy. The common practice when this option does not exist is to use the vertical values as the dipole sonic was not run in the past because of the higher cost. The majority of hydraulic fracturing treatment in the past has been done in sands and carbonate where the horizontal and vertical elastic properties are the same. However in unconventional when dealing with high clay formation the anisotropy is higher which requires an understanding of the medium. The fracture propagates transversely. This implies that the fracture opens against the horizontal elastic properties. Meaning we need to use horizontal elastic properties in the design process.
  2. The PKN used neglects the effect of DFIT. This can be a valid point as the permeability of the formation is very low leading to a negligible leakoff. This will result in close geometry to the case where leakoff is accounted for. Note that even if we have a higher leakoff. The net pressure would be lower than the estimated one. Meaning that the argument that we should have lower net pressure is still valid in both cases.

The plots were fixed. And the longer fractures were discussed.

Round 2

Reviewer 1 Report

The response is reasonable. I would suggest not using the word 'anisotropic' in the title. In mechanics, anisotropy has a well-defined meaning, i.e., the elastic deformation behavior can not be characterized simply by two constants, such as young's modulus and poison ratio. I accepted reviewing this paper mainly because I thought the layers were treated as anisotropic elastic solids.

Author Response

Thank you so much for the recommendation. the title was fixed and corrected, I really appreciate your feedback 

Reviewer 3 Report

Authors’ responses are fine. Authors need to make sure that all cited articles/papers within the text are listed in the reference list, particularly, new cited articles in Lines 33-34 and 56.

 With regard to Lines 40-42, definitely natural fractures play a great role in fracture geometry growth and finally production, when multiple transverse fractures in a horizontal well in unconventional formation intersect natural fractures. Another relevant paper may add value, if the authors think so. 

 With regard to one comment of fracture design optimization (Sec 4.3), I think the whole study would help the fracture designer estimate some mechanical properties, which are required in fracture design optimization algorithm.

 Authors need to look at the English language once again, wherever possible to polish the language.

Finally, this is a very good work, which I really appreciate in this effort. This will definitely interest the academia and industry.

Author Response

Thank you so much for the time you spent reviewing the paper, all comments has been addressed and the papers fixed; thank you so much